# A Close Relationship Between Ultra-Processed Foods and Adiposity in Adults in Southern Italy

**DOI:** 10.3390/nu16223923

**Published:** 2024-11-17

**Authors:** Mariana Di Lorenzo, Laura Aurino, Mauro Cataldi, Nunzia Cacciapuoti, Mariastella Di Lauro, Maria Serena Lonardo, Claudia Gautiero, Bruna Guida

**Affiliations:** 1Department of Clinical Medicine and Surgery, Physiology Nutrition Unit, University of Naples Federico II, Via Sergio Pansini 5, 80131 Napoli, Italy; marianadilorenzo.mdl@gmail.com (M.D.L.); laura.aurino@gmail.com (L.A.); nunzia_cacciapuoti@libero.it (N.C.); msdl@hotmail.it (M.D.L.); mslonardo91@gmail.com (M.S.L.); claudia.gautiero@gmail.com (C.G.); bguida@unina.it (B.G.); 2Division of Pharmacology, Department of Neuroscience, Reproductive Sciences and Dentistry, University of Naples Federico II, 80131 Naples, Italy

**Keywords:** ultra-processed foods, soft drinks, obesity, BMI, visceral adiposity

## Abstract

Background/Objectives: One of the main culprits of the obesity epidemic is the obesogenic food environment, which promotes the consumption of ultra-processed foods (UPFs) that are highly palatable, have low nutritional quality and a high caloric impact and are economical and ready to use. This monocentric retrospective study explored the association between UPFs, obesity and adiposity measurements among adults living with obesity in Southern Italy. Methods: According to their Body Mass Index (BMI) values, 175 participants (63M) were recruited and stratified into three groups. To evaluate their usual eating habits, PREDIMED and the Nova Food Frequency Questionnaire (NFFQ) were administered to investigate Mediterranean diet (MD) adherence and UPF consumption. Anthropometric and biochemical measurements, body composition, as well as visceral obesity indices were collected. Results: The data showed an increase in UPF consumption as the BMI increased, with a concomitant decrease in MD adherence. Soft drinks were the most representative UPF in all groups, and we observed a significant increase in such consumption as the BMI increased. In addition, in the highest tertile of UPF consumption, there was an increase in adiposity indices. Conclusions: Our data suggest that high UPF consumption correlates with an increased BMI and visceral adiposity, and it is a predictive risk factor for the occurrence of non-communicable diseases.

## 1. Introduction

Obesity represents the most common form of malnutrition in the majority of countries and is one of the greatest public health challenges worldwide [1]. Among the several factors involved in the genesis of overweight and obesity, food processing is in the eye of the storm as an emerging line of investigation [2,3]. Global food systems have undergone profound change in terms of technology and food processing, and these increasingly sophisticated processing methods not only achieve very high food safety standards but also alter the structure, nutritional profile and taste of food, making it more palatable [2,4,5]. This phenomenon has caused a change in eating styles and habits in modern society across the globe, leading to the “Westernized diet”, with a massive increase in the consumption of various industrial highly processed and energy-dense foods [6]. Among these foods, we find ultra-processed foods (UPFs), a term coined by Monteiro’s research group who have introduced the NOVA (not an acronym) food classification system [7]. This international classification is widely used in the scientific literature and assumes that the impact on health is mainly determined by the degree of transformation that the food has undergone while disregarding other aspects such as the nutritional profile, energy and nutrient supply or contribution in terms of biologically active minor components [7]. The NOVA classification identifies four food groups: non-processed or minimally processed foods (MPFs), processed culinary ingredients (PCIs), processed foods (PFs) and ultra-processed foods (UPFs) [8]. Of interest for the present study, according to the NOVA system, UPFs are defined as manufactured products that are predominantly made of industrial ingredients with little or no whole foods [6]. They share a broad range of peculiarities: they are highly palatable, ready to eat and economical and have attractive packaging, as well as being supported by multi-media and other aggressive advertising campaigns. From a nutritional point of view, these foods are energy-dense foods, with high amounts of sugars, salt, saturated fats, trans fats, additives such as emulsifiers and preservatives and low in vitamins, fibers and other essential micronutrients [6,9]. Examples of typical UPFs are soft or energy drinks, packaged snacks, meal substitutes, poultry and fish nuggets, sausages, hot dogs and other reconstituted meat products, breakfast cereals and energy bars, candy, cookies, ice-creams or alcoholic drinks such as whisky, gin, rum or vodka [6]. The NOVA classification has been used extensively in published papers to investigate the association between UPF consumption and different health outcomes. In recent decades, numerous observational studies and meta-analyses have reported a linear relationship between UPF intake and worsened nutritional status, with increased risks of weight gain, overweight and obesity [9,10,11,12,13]. It is also known that an increased intake of UPFs can replace unprocessed foods and freshly prepared meals [8], which are the basis of traditional dietary models such as the Mediterranean diet (MD), which is recognized to promote longevity [14]. The relationship between UPFs and health is a source of heated debate by the scientific community, as the evaluation of this association presents methodological limitations. First, the food consumption data analyzed in existing studies were obtained with survey tools such as food frequency questionnaires (FFQs), food diaries and 24 h recalls, which were not designed to assess food processing, and this leads to a heterogeneity of evaluation and reduces the accuracy of the results [15]. In addition, most studies have only considered the Body Mass Index (BMI) for the evaluation of obesity [8,9,16,17]. Despite BMI being the most widely used parameter for assessing obesity [18,19], it has important limitations in terms of the evaluation of the quantity and distribution of body fat, and thus, it does not reflect the impact of excess adiposity on health and does not allow for a complete understanding of the pathophysiology of the disease. In light of the above data, the purpose of the present study was to assess UPF consumption in relation to the degree of obesity, using validated instruments that are specifically designed for this purpose, and the possible relationship with adherence to the MD. Furthermore, a secondary goal was to evaluate the possible impact of UPFs on adiposity using visceral adiposity indices as parameters.

## 2. Materials and Methods

### 2.1. Study Design and Data Collection

This monocentric retrospective study was approved by the Ethical Committee of the Federico II University Medical School of Naples on 10 February 2022 (Project identification code 481/21), and written informed consent was obtained from each participant.

A total of 175 participants (63 M) attending the Outpatients Clinic of the I.P. “Diet Therapy in transplantation, renal failure and chronic pathology”, University of Naples Federico II, between October and December 2021 were recruited and stratified according to the WHO BMI classification for the Caucasian population [18] into three groups:-Group 1: Obesity, Class I, BMI 30–34.9 kg/m^2^;-Group 2: Obesity, Class II, BMI 35–39.9 kg/m^2^;-Group 3: Obesity, Class III, BMI ≥40 kg/m^2^.

Recruitment took into account the following inclusion criteria:-Age between 18 and 65 years old;-BMI ≥30 kg/m^2^.

Individuals who were bedridden or suffered from diabetes, cancer, dementia, depression or neurological disorders, as well as pregnant and lactating women, were excluded from the study population.

Data about age, gender, marital status, education level, occupation, place of residence and physical activity were collected. Marital status was categorized as unmarried/single, married/partner and divorced. Education level was categorized as secondary school or below, high school and university. Occupation was classified as unemployed and employed, and place of residence was separated into metropolis and small and medium-sized cities. Self-reported data about physical activity (sedentary, medium, heavy) were recorded.

### 2.2. Anthropometric Measurements and Body Composition Analysis

Nutritional status was assessed using anthropometric measurements, and for this purpose, all subjects had to be in light clothes and without shoes. Body weight and height were determined using a calibrated balance beam scale and a stadiometer (Seca 711; Seca Hamburg, Germany); then, the BMI was calculated. Waist circumference (WC) in centimeters was assessed, according to the National Institutes of Health (NIH) protocols, with a no-stretch tape measure halfway between the lower edge of the rib cage and the iliac crest, as previously reported [20]. To assess body composition, bioelectrical impedance analysis (BIA) parameters such as fat-free mass (FFM), fat mass (FM) (expressed in %) and FM index (FMI) were detected using a tetrapolar BIA (RJL 101; Akern SRL, Florence, Italy) with an 800 μA current at a single frequency of 50 kHz [21]. To correctly perform the bioimpedance analysis, all enrolled subjects were asked not to consume caffeine or alcohol and not to carry out physical activity for at least 24 h and to fast at least 6–12 h before the outpatient visit. In addition, all subjects were invited to remove metal from their clothing and body and empty their bladder before measurement.

This analysis was preceded by positioning the subject in a supine position and performed after a wait of approximately 10′ to allow for a homogeneous distribution of body fluids. Since all study subjects were living with obesity, we placed rolled paper between the arms and trunk and, if necessary, between the legs to separate the limbs. Briefly, after cleaning the skin surface with alcohol, ensuring that the subjects had not used lotions or oils before measurement, the 4 electrodes were applied by positioning them at ≥5 cm and ≥7.5 cm, respectively, on the hand and foot using the respective tweezers, and the analysis was carried out [22].

### 2.3. Biochemical and Clinical Parameters

Overnight fasting venous blood samples were collected from enrolled participants, and blood glucose (Gly), insulin, total cholesterol (Tot-C), HDL cholesterol (HDL-C), LDL cholesterol (LDL-C), triglycerides (TGs) and uric acid were evaluated by using standard analytic laboratory methods. Shortly, TG/HDL-C ratio was computed to detect subjects with insulin resistance (IR) [23,24]. Furthermore, all subjects were investigated for hypertension according to the 2021 ESC guidelines [25]. Subjects were asked to not consume caffeine or food and not to smoke for at least 30′. Blood pressure measurement was performed as follows: a common aneroid sphygmomanometer was used with the cuff size adapted to the subject’s arm. Subjects remained seated for 5′ with their back firmly supported on the chair, legs uncrossed, feet flat and both arms resting on the table, with their center at heart level. The cuff was wrapped around the arm so that it was snug but not tight, approximately 2 cm above the elbow crease. We identified the radial artery at the wrist, and the stethoscope was positioned over the humeral artery, while the cuff was inflated to approximately 30 mmHg above the value at which the radial pulse disappeared. The cuff was deflated at a rate of 2–3 mmHg per second. The first sound heard (1st Korotkov tone) corresponded to the Systolic Pressure, while the Diastolic Pressure corresponded to the last sound heard. The procedure was repeated 3 times at 1 min apart [26].

### 2.4. Nutritional Assessments

At baseline, to evaluate eating habits, the validated PREvención con DIetaMEDiterránea (PREDIMED) and Nova Food Frequency Questionnaire (NFFQ) were administered to each recruited subject by a qualified nutritionist during a face-to-face interview verifying in real time the correspondence of answers to the items common to both questionnaires. The PREDIMED questionnaire consists of 14 items concerning the consumption of the staple foods of the MD and, by assigning a score of 1 and 0 for each item, the PREDIMED score was briefly calculated as follows: 0–5, lowest adherence; 6–9, average adherence; ≥10, highest adherence [27,28]. The NFFQ was designed to estimate the intake (g/day) and the weight ratio (%) of the NOVA food groups in Italian adults [28] and consists of 94 items with relative frequency and portion size of consumption. Participants were asked to answer, taking into account their diet on a typical month over the past 12 months, and daily food consumption was estimated by multiplying the portion size by the consumption frequency for each food item [28]. When processing the NFFQ data, PCIs were grouped with PFs, as these products are not intended to be consumed alone as food but are usually used to prepare other foods [28,29]. Moreover, the proportion of UPFs was categorized into tertiles, corresponding to low (first tertile, T1), medium (second tertile, T2) and high (third tertile, T3) consumption. Further, a photographic food atlas (≈1000 photographs) of known portion sizes was used to better quantify food and drink portions [30], and the average daily calorie intake was calculated using a dedicated software [The WinFood software (version 3.9), Medimatica Srl, Colonella, TE, Italy].

### 2.5. Indicators of Adiposity

To better characterize the possible presence of visceral adiposity, the following adiposity indices were calculated:-Waist-to-height ratio (WHtR) [31]:WHtR: WC/height (both expressed in centimeters).-Visceral Adiposity Index (VAI) [32]:VAI (men): (WC)/[39.68 + (1.88 × BMI)] × (TG/1.03) × (1.31/HDL);VAI (women): (WC)/[36.58 + (1.89 × BMI)] × (TG/0.81) × (1.52/HDL).-Lipid Accumulation Product (LAP) [33]:LAP (men): (WC − 65) × TG;LAP (women): (WC − 58) × TG.-Cardiometabolic Index (CMI) [34]:CMI: (TG/HDL) × WHtR.-Waist Triglyceride Index (WTI) [34]:WTI: WC × TG.

### 2.6. Statistical Analysis

Statistical analysis was performed using IBM Corp Released 2021, IBM SPSS Statistics for Windows, Version 28.0. Armonk, NY, USA: IBM Corp. For categorical variables, absolute numbers and frequencies (%) are shown.

The Kolmogorov–Smirnoff test was used to check normality. The Levene test was used to evaluate the equality of variance. Normally distributed variables were given as the mean ± standard deviation (SD), and Chi-squared test and analysis of variance (ANOVA) were performed. Non-normally distributed variables were expressed as the median and interquartile range, and a nonparametric test of multiple comparison of Krustal–Wallis was carried out. The statistical significance was set at *p* < 0.05.

## 3. Results

### 3.1. Assessment of Eating Habits in Relation to BMI

A total of 175 participants (36% males) with a mean age of 43.3 ± 12.6 years and a BMI of 42 (95% CI: 37–47) were included in this analysis. In Table 1, the baseline socio-demographic features, anthropometric measures and body composition characteristics of the study population according to the BMI are reported. Overall, more than half of the participants had attended high school and were married. No significant differences were observed for age, gender, educational level, occupation, marital status or physical activity among the groups. In addition, participants with higher BMI values lived in small and medium-size cities compared to those in Group 1, who predominantly resided in a metropolis (*p* < 0.05). The biochemical and clinical parameters of the study population according to their BMI values are reported in Table 2.

The participants’ eating habits according to their BMI values are listed in Table 3.

As shown in Table 3, the consumption of UPFs tends to increase significantly with an increasing BMI. In detail, in terms of daily consumption, in Group 1, the percentage of UPF consumption was 18.2% (95% CI: 16.7–23.5), in Group 2, it was 18.6% (95% CI: 20–27.6), and in Group 3, it was 26.2% (95% CI%: 26.6–32.5) (*p* < 0.01) (Table 3). The amount of UPFs in the diet corresponded to 274.2 (95% CI: 241.5–495.5) g/day in Group 1, 278.4 (95% CI: 312.2–514) g/day in Group 2 and 526.2 (95% CI: 575.9–802.8) g/day in Group 3 (*p* < 0.001) (Table 3). Interestingly, this increase in UPF consumption occurred without a significant change in daily calorie intake between groups (Table 3).

From the total amount of UPF consumed, soft drinks were found to be the main contributors in all BMI groups considered, and, again, it was observed that the consumption of such drinks tended to increase significantly as the BMI increased: in fact, in Group 3, this consumption was found to be more than doubled compared to Group 1. Specifically, in Group 1, the proportion of soft drink consumption was 11.7% (95% CI: 7.7–25.3), in Group 2, it was 19.7% (95% CI: 17–30.1), and in Group 3, it was 27% (95% CI: 25.9–36.5) (*p* < 0.01) (Table 3). In addition, in Group 3, compared to both Group 1 and Group 2, significant increases in the g/day consumption of sweet and savory snacks (*p* < 0.05), ice-cream (*p* < 0.05), chips and French fries (*p* < 0.01), sausages and würstel (*p* < 0.01) and nuggets and sticks (*p* < 0.001) were observed (Table 3). In terms of daily intake, no significant differences were observed either in g/day or weight ratio (%) between groups regarding the consumption of energy drinks, alcohol, packaged breads, buns, biscuits, chocolate and fish sticks (Table 3).

In addition to UPF consumption, we also calculated the consumption of unprocessed or minimally processed (MPF) and processed (PF + PCI) foods. With regard to MPFs, a significant reduction in their consumption was observed as the BMI increased (*p* < 0.001): indeed, consumption went from 55.1% (95% CI: 50.1–58.8) in Group 1 to 49.8% (95% CI: 43.7–52.2) and 43.3% (95% CI: 40.7–46.6) in Groups 2 and 3, respectively (Table 3).

On the other hand, no significant difference was found with regard to the daily percentage consumption of processed food (PF + PCI) (Table 3).

A high adherence to the Mediterranean diet was not observed in any of the three groups, and a significant reduction was observed as the BMI increased (*p* < 0.005). The PREDIMED score in Group 3 was 5 (95% CI: 4.8–5.3), showing poor adherence to the MD compared to Group 1 and Group 2 (Table 3).

### 3.2. Socio-Demographic Characteristics and Adiposity Indices Across UPF Tertiles

Further analysis was conducted by dividing the study population into tertiles of UPFs as follows:-T1: % UPF < 18.3;-T2: % UPF ≥ 18.3 and <29.1;-T3: % UPF ≥ 29.1.

Notably, a significant linear trend between participants’ age and UPF consumption was observed: indeed, in the highest tertile (T3), the participants were younger compared to both T1 and T2 (*p* < 0.05) (Table 4). As shown in Table 4, no significant variation was observed between the tertile groups regarding physical activity and the other socio-demographic characteristics considered.

Interestingly, the data obtained show that higher UPF consumption is associated with higher levels of all adiposity indices considered (Table 4). In detail, WHtR went from 0.6 (95% CI: 0.6–07) in T1 to 0.7 (95% CI: 0.7–08) and 0.7 (95% CI: 0.6–0.8) in T2 and T3, respectively (*p* < 0.05) (Table 4).

Similarly, in the highest tertile of UPF, significant increases in both VAI and LAP were observed, with values rising from 1.4 (95% CI: 1.2–1.7) and 80.4 (95% CI: 65.9–91.4) in T1 to 2 (95% CI: 1.7–2.7) and 113.6 (95.3–129) (*p* < 0.001) in T3, respectively (Table 4). Significant increases in the CMI (*p* < 0.001) and WTI (*p* < 0.001) were also observed in T3 compared to T1 and T2 (Table 4).

## 4. Discussion

The present study shed light on the positive association between a high UPF intake and the degree of obesity and adiposity indices in a cohort of adults from Southern Italy. Several studies have already investigated the relationship between UPFs and obesity but, to the best of our knowledge, the present study is one of the few to explore the association between UPF consumption and adiposity using specifically formulated questionnaires such as the NFFQ.

According to previously published evidence obtained both in children, adolescents [35,36] and adults [37,38,39], our findings support the evidence of a positive association between UPF consumption and weight gain. In fact, it has been observed that as the BMI increases, UPF consumption increases, while adherence to the MD decreases, with a reduced consumption of fresh and minimally processed foods such as typical products of the MD, such as fruit, vegetables and fish.

Various potential mechanistic explanations for this relation have been postulated. First of all, as mentioned before, UPF consumption is predictive of low-quality diets; therefore, a high consumption of UPFs is linked to a higher energy intake [40,41] with an associated greater intake of simple sugars and saturated fats and a lower intake of proteins, fibers and micronutrients and vitamins [39,42]. The high energy density, hyper-palatability and low satiating power of these foods can easily lead to overeating [43,44]. Specifically, it has been shown that UPFs can trigger addictive eating behavior by modifying the reward mechanism in the mesolimbic circuits of the dopamine, taste and oral somatosensory brain regions, contributing to overeating [45]. In addition, industrial processing alters food matrices by reducing the duration of chewing and the gastric emptying time and accelerating the digestion and absorption of nutrients [46,47,48]. Additionally, it has been suggested that processed foods are more thermodynamically efficient than the comparable non-processed or minimally processed foods and thereby confer a metabolic disadvantage in relation to obesity [49]. Differences in quality between MPFs and UPFs help to explain why more energy is required in the digestion of unprocessed or minimally processed foods [50]. In addition, deep processing and food packaging materials can lead to the formation of potentially toxic compounds and contaminants such as acrolein, furans, polycyclic aromatic hydrocarbons, phthalates and bisphenols [51,52,53,54]. These compounds play an important role in the development of obesity and obesity-related diseases [55], as they can interfere with insulin sensitivity and glucose tolerance [56] and are capable of promoting adipogenesis by controlling and/or fostering lipid accumulation [57] or the energy balance, favoring food conservation and overeating and thus altering the hormonal control of the hunger–satiety circuit [56]. Furthermore, UPFs contain higher amounts of advanced glycation end products (AGEs) [58]. AGEs and their respective precursors are produced in food production during dry and high-temperature cooking methods (frying, baking, steaming). These molecules, when excessive, can promote renal structure damage, oxidative stress, hyperglycemia, hyperlipidemia and endothelial dysfunction [59,60].

Our data highlighted that among UPFs, consumption of soft drinks was the most consistent in all BMI groups considered, and more precisely, this consumption heightened as the degree of obesity increased. Soft drinks such as common carbonated drinks are a class of water-based non-alcoholic beverages usually containing artificial sweeteners [61]. Artificial food additives, such as sweeteners, emulsifiers and preservatives, can stimulate appetite and cause sugar cravings and alter the gut microbiota composition, potentially resulting in metabolic abnormalities and promoting obesity [62,63]. Therefore, the consumption of soft drinks may lead to weight gain either directly through increased energy intake from the drinks themselves or indirectly due to their reduced satiating power [64,65], which leads to an increased intake of foods [66]. A growing number of studies have shown that habitual consumption of ultra-processed beverages such as soft drinks is linked to metabolic syndrome and increased cardiovascular risk [67], both in adolescents and adults [68,69], as it is associated with an increase in visceral adipose tissue and a greater deposition of ectopic fat, mainly in the liver and muscles [70].

In line with other studies [29,71], the other main food categories that contributed to UPF intake in our study population were ready-to-eat and heated meals such as sweet and savory snacks, chips and French fries, sausages and other reconstituted meat products. In order to improve their taste and durability, these fast foods are typically overloaded with sodium and other phosphate-based additives [72] which, similarly to other artificial food additives, may be obesogenic [73,74].

In this context, our results emphasized an increase in adiposity indices with increasing UPF consumption. Indeed, participants in the highest tertile of UPF consumption presented with higher levels of adiposity indices. Indices of adiposity such as WHtR, VAI, LAP, CMI and WTI combine anthropometric and biochemical data and are useful parameters for assessing visceral adipose tissue and a better correlation with metabolic conditions such as insulin resistance, diabetes, metabolic syndrome, non-alcoholic liver steatosis and cardiovascular disorders [32,33,75]. In this regard, previous observations have shown that the WHtR, VAI and CMI indexes have high sensitivity and specificity for detecting metabolic syndrome in children, adolescent and adults with obesity [76,77]. Recent interesting evidence has shown that WHtR has a sensitivity of 100% compared to FM% assessed with dual-energy X-ray absorptiometry (DEXA) as the gold standard [78], which supports the validity of using adiposity indices for the early assessment of cardiometabolic outcomes.

This study is one of the first to explore the association between adiposity and UPFs in a representative national sample using a validated questionnaire specifically designed to estimate food and beverage intake according to the NOVA classification. Nonetheless, some limitations should be considered. First, this is a retrospective study that might be subject to bias; thus, multicentric prospective studies on larger populations with multiple regression analyses are needed to confirm our preliminary findings. Second, this study could benefit from validated questionnaires to assess sleep quality and stress levels to add depth to the analysis.

## 5. Conclusions

In conclusion, the results of the present study pointed out that UPF consumptions may jeopardize human health. From a nutritional point of view, the main problems with UPFs are their low nutritional profile, poor quality and low satiating power, in addition to their ability to interfere with the reward circuit, activating a vicious circle that leads to overeating. From a health point of view, a high consumption of these products has been seen to be predictive of low diet quality and increased visceral adiposity, predisposing individuals to a greater risk of developing cardiovascular diseases. These results highlight the importance of implementing public health strategies to improve population health by promoting the Mediterranean diet and limiting UPF intake in favor of higher-quality products.

## Figures and Tables

**Table 1 nutrients-16-03923-t001:** Socio-demographic, anthropometric and body composition characteristics of population.

	All Patients *n* = 175	Group 1 *n* = 26	Group 2 *n* = 48	Group 3 *n* = 101	*p*-Value
Age, Years *	43.3 ± 12.6	41.6 ± 11.8	42.2 ± 12.4	43.8 ± 12.9	0.6
Male, *n* (%)	63 (36%)	7 (26.9%)	12 (25%)	44 (43.6%)	0.06
Marital status, *n* (%)					0.91
* Unmarried/Single*	60 (34.3%)	7 (26.9%)	17 (35.4%)	36 (35.6%)	
* Married/partner*	97 (55.4%)	16 (61.5%)	27 (56.3%)	54 (53.5%)	
* Divorced*	18 (10.3%)	3 (11.5%)	4 (8.3%)	11 (10.9)	
Education level, *n* (%)					0.24
* Secondary school or below*	45 (25.7%)	4 (15.4%)	10 (20.8%)	31 (30.7%)	
* High school*	110 (62.9%)	17 (65.4%)	31 (64.6%)	62 (61.4%)	
* University*	20 (11.4%)	5 (19.2%)	7 (14.6%)	8 (7.9%)	
Occupation, *n* (%)					0.25
* * *Unemployed*	86 (49.1%)	11 (42.3%)	20 (41.7%)	55 (54.5%)	
* * *Employed*	89 (50.9%)	15 (57.7%)	28 (58.3%)	46 (45.5%)	
Place of residence, *n* (%)					*<0.05*
* * *Metropolis*	98 (56%)	20 (76.9%)	28 (58.3%)	50 (49.5%)	
* * *Small and medium-sized cities*	77 (44%)	6 (23.1%)	20 (41.7%)	51 (50.5%)	
Physical activity level, *n* (%)					0.88
* * *Sedentary*	166 (94.8%)	25 (96.2%)	45 (93.7%)	96 (95%)	
* * *Medium*	8 (4.6%)	1 (3.8%)	3 (6.3%)	4 (4%)	
* * *Heavy*	1 (0.6%)			1 (1%)	
BW, kg **	112.9 (97.9–138.2)	85.1 (78.2–95.1)	100 (92.5–109.4)	129.1 (114.9–150.1)	*<0.001*
BMI, kg/m^2^ **	42 (37–47)	33 (31.6–33.8)	37.6 (36.3–38.7)	46.2 (43.2–52.8)	*<0.001*
WC, cm **	113 (103.7–129)	97.5 (93.2–104)	105.5 (98–111)	125 (116.2–139)	*<0.001*
FFM, % *	55.9 ± 8.3	55.5 ± 9.7	57.3 ± 8.5	55.3 ± 7.8	0.4
FM, % *	49.7 ± 10.4	46.1 ± 10.3	48.6 ± 10.7	51.1 ± 10.2	0.07
FMI, kg/m^2^ **	18.2 (14.2–22.7)	11.9 (10.6–13.3)	16 (13.7–17.6)	22.2 (18.8–25.2)	*<0.001*

Continuous variables are expressed as * mean ± standard deviation (SD) or ** median and interquartile range (IQR). Categorical variables are expressed as numbers and percentages. Abbreviations: BW, Body Weight; BMI, Body Mass Index; WC, Waist Circumference; FFM, fat-free mass; FM, fat mass; FMI, Fat Mass Index.

**Table 2 nutrients-16-03923-t002:** Biochemical and clinical parameters of study population.

	All Patients *n* = 175	Group 1 *n* = 26	Group 2 *n* = 48	Group 3 *n* = 101	*p*-Value
Glucose, mg/dL **	93 (85–103.2)	87 (80–96)	91 (82–103)	95.5 (88–105.7)	*<0.05*
Insulin, µU/mL **	16.7 (10.8–30.3)	11.1 (8.1–21)	14 (8.9–19.1)	18.3 (13.1–35.2)	*<0.001*
Tot-C, mg/dL *	183.2 ± 34	179.9 ± 35.5	189.6 ± 39.6	181.3 ± 30.8	0.4
LDL-C, mg/dL **	114 (89.2–136)	101 (83–126)	111.3 (93–128)	119 (93.2–143)	0.2
HDL-C, mg/dL **	50 (40.2–59)	56.8 (41.9–64)	54 (45–63)	46 (39–51)	*<0.01*
TG, mg/dL **	118 (88.5–163.5)	111 (81–155)	103.5 (78–171.5)	123 (93.2–161.5)	0.4
TG/HDL ratio **	2.4 (1.7–3.5)	2.1 (1.6–2.6)	2 (1.3–3.7)	2.6 (1.9–3.7)	*<0.01*
Uric acid, mg/dL *	5.5 ± 1.4	4.9 ± 0.8	5.1 ± 1.7	5.9 ± 1.4	*<0.01*
SBP, mmHg **	125 (120–140)	120 (117.5–130)	120 (120–133.7)	130 (120–140)	*<0.05*
DBP, mmHg **	80 (80–90)	80 (78.7–80)	80 (76.2–80)	80 (80–90)	*<0.001*

Data are expressed as * mean ± standard deviation (SD) or ** median and interquartile range (IQR). Abbreviations: Tot-C, total cholesterol; LDL-C, Low-Density Lipoprotein–cholesterol; HDL-C, High-Density Lipoprotein–cholesterol; TG, triglyceride; SBP, Systolic Blood Pressure; DBP, Diastolic Blood Pressure.

**Table 3 nutrients-16-03923-t003:** Eating habits according to BMI groups.

	All Patients *n* = 175	Group 1 *n* = 26	Group 2 *n* = 48	Group 3 *n* = 101	*p*-Value
Kcal/day *	3632.5 (3069.7–4388.7)	3714 (2959–4202.2)	3271 (3012.5–3987.7)	3741 (3190.5–4598.5)	0.07
MPF g/day *	827.8 (626.5–1050.7)	914.8 (692.1–1140.3)	789.5 (497–966.4)	814.8 (635.5–1067.1)	0.1
% of tot food *	47.2 (37.2–58.1)	55.1 (48–62.4)	49.8 (38.6–60.7)	43.3 (31.3–55.7)	*<0.001*
PF (PF + PCI) g/day *	461.5 (358.9–587.1)	398.1 (297.3–489.6)	424.8 (326.1–555.7)	510.5 (390.8–693.6)	*<0.001*
% of tot food *	25.6 (21.1–32.4)	24.4 (19.4–29.9)	26.1 (22.2–33.1)	25.6 (20.1–32.3)	0.4
UPF g/day *	394 (255–628.5)	274.2 (212–397.8)	278.4 (224–475.4)	526.2 (329.7–799.4)	*<0.001*
% of tot food *	23.1 (16–34.7)	18.2 (14.2–25.6)	19 (15–30)	26.2 (19.1–37.8)	*<0.01*
*UPF Subgroups*					
*Energy Drinks* g/day *	0 (0–20)	0 (0–5)	0 (0–0)	0 (0–28.5)	0.3
% of tot UPFs *	0 (0–2.3)	0 (0–1.6)	0 (0–0)	0 (0–4)	0.5
*Soft Drinks* g/day *	48.5 (0–200)	28.2 (0–47.8)	47.1 (0–141.4)	100 (33–330)	*<0.01*
% of tot UPFs *	21.7 (0–43.9)	11.7 (0–20.7)	19.7 (0–35)	27 (6.8–49)	*<0.01*
*Alcohol* g/day *	0 (0–0)	0 (0–0)	0 (0–0)	0 (0–0)	0.9
% of tot UPFs *	0 (0–0)	0 (0–0)	0 (0–0)	0 (0–0)	0.3
*Packaged breads* g/day *	5 (0–15)	6 (0–16.6)	5 (0–14.2)	7.1 (0–21.4)	0.5
% of tot UPFs *	1.1 (0–3.9)	2.5 (0–5.6)	0.4 (0–4.4)	1.1 (0–3.4)	0.6
*Buns* g/day *	10 (0–25.7)	12.8 (0–30)	8.5 (0–20.6)	10 (3.6–25.7)	0.6
% of tot UPFs *	2.2 (0–6.1)	4.2 (0–12.2)	2.8 (0–6.2)	1.9 (0.6–5.1)	0.3
*Sweet/savory snacks* g/day *	9.5 (3–25.7)	8.5 (0–18.2)	6.2 (0–16.6)	12.8 (3–27.8)	*<0.05*
% of tot UPFs *	2.5 (0.1–4.9)	2.8 (0–5.1)	2 (0–4.2)	2.5 (0.5–5.5)	0.4
*Snacks* g/day *	14.2 (5–42.8)	7.1 (3.7–37.5)	18.2 (5–28.5)	14.2 (5–50)	0.4
% of tot UPFs *	3.4 (0.4–9.1)	3 (1.1–11.4)	3.8 (0–9.6)	3 (0.5–8.1)	0.8
*Biscuits* g/day *	17.1 (4.2–38.5)	17.1 (2.2–26.7)	15 (4.8–45)	21.4 (7.2–40)	0.3
% of tot UPFs *	4.7 (0.9–8.1)	5.1 (0.7–7.5)	5.2 (0.9–11.2)	4.2 (0.9–7.9)	0.8
*Ice-cream* g/day *	0 (0–10)	0 (0–0)	0 (0–10)	0 (0–10.2)	*<0.05*
% of tot UPFs *	0 (0–2)	0 (0–0)	0 (0–1.9)	0 (0–2.3)	*<0.05*
*Chocolate* g/day *	7.1 (0–20)	8.5 (0–17.8)	8.5 (0–20)	6.4 (0–20.5)	0.9
% of tot UPFs *	1.6 (0–3.9)	3.1 (0–5.6)	1.9 (0–5.6)	1.2 (0–3.4)	0.2
*Chips and French fries*, g/day *	15 (0–32.1)	0 (0–21.4)	0 (0–28)	21.4 (0–42.8)	*<0.01*
% of tot UPFs *	2.1 (0–8)	0 (0–6.8)	0 (0–8.2)	2.8 (0–8.5)	0.1
*Sausages and würstel* g/day *	20 (10–35)	14.2 (7.7–22.8)	14.2 (0–26.7)	28.5 (14.2–42.8)	*<0.01*
% of tot UPFs *	4.5 (1.2–8.2)	4 (1.2–7.1)	4 (0–8.7)	4.9 (1.3–8.3)	0.7
*Nuggets and sticks* g/day *	14.2 (0–28)	5 (0–14.4)	0 (0–14.8)	14.2 (0–28.5)	*<0.001*
% of tot UPFs *	2.1 (0–5.7)	0.7 (0–5.6)	0 (0–6)	3.2 (0–5.6)	0.1
*Fish sticks* g/day *	10 (0–15)	5 (0–16.6)	0 (0–18.7)	10 (0–17.5)	0.8
% of tot UPFs *	1.4 (0–4.5)	1.5 (0–6.8)	1.4 (0–5.7)	1.4 (0–3.9)	0.5
**PREDIMED SCORE** *	5 (5–6)	6 (5–6.5)	6 (5–6)	5 (4–6)	*<0.01*

Data are expressed as * median and interquartile range (IQR). Abbreviations: MPF, non-processed or minimally processed food; PF, processed food; PCI, processed culinary ingredient; UPF, ultra-processed food.

**Table 4 nutrients-16-03923-t004:** Socio-demographic characteristics and adiposity indices across UPF tertiles.

Percentage of UPF ^1^	Tertile 1 *n* = 57 <18.3	Tertile 2 *n* = 59 18.3 ≤ % < 29.1	Tertile 3 *n* = 59 ≥29.1	*p*-Value
Age, Years *	46.1 ± 12.01	43.5 ± 11.2	39.8 ± 13.7	*<0.05*
Male, *n* (%)	24 (42.1%)	17 (28.8%)	22 (37.3%)	0.31
Marital status, *n* (%)				0.38
* Unmarried/Single*	16 (28.1%)	20 (33.9%)	24 (40.7%)	
* Married/partner*	35 (61.4%)	35 (59.3%)	27 (45.7%)	
* Divorced*	6 (10.5%)	4 (6.8%)	8 (13.6%)	
Education level, *n* (%)				0.06
* Secondary school or below*	17 (29.8%)	16 (27.1%)	12 (20.3%)	
* High school*	29 (50.9%)	37 (62.7%)	44 (74.6%)	
* University*	11 (19.3%)	6 (10.2%)	3 (5.1%)	
Occupation, *n* (%)				0.81
* * *Unemployed*	30 (52.6%)	31 (52.5%)	28 (47.5%)	
* * *Employed*	27 (47.4%)	28 (47.5%)	31 (52.5%)	
Place of residence, *n* (%)				0.051
* * *Metropolis*	33 (57.9%)	39 (66.1%)	26 (44.1%)	
* * *Small and medium-sized cities*	24 (42.1%)	20 (33.9%)	33 (55.9%)	
Physical activity level, *n* (%)				0.41
* * *Sedentary*	53 (92.9%)	55 (93.2%)	58 (98.3%)	
* * *Medium*	3 (5.3%)	4 (6.8%)	1 (1.7%)	
* * *Heavy*	1 (1.8%)			
WHtR **	0.6 (0.6–0.7)	0.7 (0.6–0.7)	0.7 (0.6–0.8)	*<0.05*
CMI **	0.7 (0.4–0.8)	1 (0.5–1.1)	1.2 (0.6–1.3)	*<0.001*
VAI **	1.4 (1–1.9)	1.9 (1.3–2.4)	2 (1.4–2–6)	*<0.01*
LAP **	80.4 (40.7–90.6)	96.8 (51.6–102.7)	113.6 (66.3–125.4)	*<0.001*
WTI **	161.5 (97.9–180.4)	196.7 (113.8–212.9)	222.7 (145.1–258.7)	*<0.001*

Variables are expressed as * mean ± standard deviation (SD) or ** median and interquartile range (IQR). Categorical variables are expressed as numbers and percentages. ^1^ Percentage of total food intake. Abbreviations: UPF, ultra-processed food; WHtR, Waist-to-height ratio; LAP, Lipid Accumulation Product; VAI, Visceral Adiposity Index; WTI, Waist Triglyceride Index; CMI, Cardiometabolic Index.

## Data Availability

The data are stored in a database at the Department of Clinical Medicine and Surgery, Nutrition Physiology Unit, University Federico II of Naples, Naples 80131, Italy. It is available upon request to Bruna Guida, who is a co-author of this paper.

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
