# Peer review of "A Close Relationship Between Ultra-Processed Foods and Adiposity in Adults in Southern Italy"

_nutrients, 2024, doi:10.3390/nu16223923_

Round 1
Reviewer 1 Report
Comments and Suggestions for Authors
This study presents interesting insights into the relationship between ultra-processed food (UPF) consumption, body mass index (BMI), and adiposity among adults with obesity in Southern Italy.
The demographic specifics, such as location and socioeconomic background, could influence dietary habits and might not represent broader populations. A multicentric or more diverse sample could add robustness to the findings.
While the PREDIMED and Nova Food Frequency Questionnaire (NFFQ) tools are well-regarded, they rely on self-reporting, which can be prone to bias and inaccuracies. Self-reported dietary data often underestimates or misrepresents UPF intake, especially in populations with obesity, due to social desirability bias or memory lapses. This should be mentioned. Additionally, combining different tools to assess Mediterranean diet adherence and UPF consumption can lead to inconsistent assessments, as they may use different time frames or food categories.
The study categorizes soft drinks as the most representative UPFs. However, UPFs include a wide variety of foods, and a more detailed breakdown by type and frequency of UPF items might provide a clearer picture. The study could benefit from exploring the impact of specific categories of UPFs (e.g., snacks, desserts, packaged meals) on BMI and adiposity indices, which may influence consumption patterns differently.
Given the study's retrospective nature, establishing causality is challenging. Factors like physical activity, sleep, stress levels, and socioeconomic status, which influence both obesity and dietary choices, should be controlled or at least acknowledged. Without these, it is difficult to attribute increased BMI and adiposity solely to UPF consumption.
Although the study mentions several anthropometric and biochemical indices, it would be beneficial to specify which adiposity measurements were taken (e.g., waist-to-hip ratio, waist circumference, visceral fat percentage). A clear explanation of why specific measurements were chosen and how they relate to UPF intake would add depth to the analysis.
The study could strengthen its conclusions by discussing potential mechanisms by which UPF consumption might contribute to obesity and adiposity beyond caloric impact. For instance, the impact of additives, food matrix alterations, and hyper-palatability on satiety and metabolic health would provide a more comprehensive understanding of how UPFs exacerbate obesity.
Author Response
This study presents interesting insights into the relationship between ultra-processed food (UPF) consumption, body mass index (BMI), and adiposity among adults with obesity in Southern Italy.
Comment 1: The demographic specifics, such as location and socioeconomic background, could influence dietary habits and might not represent broader populations. A multicentric or more diverse sample could add robustness to the findings.
Reply to comment 1: We thank the reviewer for this advice. We have included information on the socio-demographic characteristics of the study population in Tables 1 and 4. This information has been detailed in the text, in the section ‘results’ (lines 203-210; lines 250-254).
In addition, we agree that our single-centre retrospective study is characterized by lower robustness of findings so we have added this within the study limits (lines 384-386)
Comment 2: While the PREDIMED and Nova Food Frequency Questionnaire (NFFQ) tools are well-regarded, they rely on self-reporting, which can be prone to bias and inaccuracies. Self-reported dietary data often underestimates or misrepresents UPF intake, especially in populations with obesity, due to social desirability bias or memory lapses. This should be mentioned. Additionally, combining different tools to assess Mediterranean diet adherence and UPF consumption can lead to inconsistent assessments, as they may use different time frames or food categories.
Reply to comment 2: We thank the reviewer for the suggestion. We have already specified in the original text that the PREDIMED and NOVA FFQ questionnaires were administered by qualified staff (nutritionists) during a face-to-face interview. We point out that with regard to the items common to the two questionnaires (e.g. consumption of fruit, vegetables, olive oil, sugary drinks etc.), these qualified staff verified the correspondence of the answers in real time. We have specified this in the text (lines 156-158).
Comment 3: The study categorizes soft drinks as the most representative UPFs. However, UPFs include a wide variety of foods, and a more detailed breakdown by type and frequency of UPF items might provide a clearer picture. The study could benefit from exploring the impact of specific categories of UPFs (e.g., snacks, desserts, packaged meals) on BMI and adiposity indices, which may influence consumption patterns differently.
Reply to comment 3:As requested, we have also detailed the frequency of consumption of each specific category of UPFs (e.g. snacks, sausages, nuggets, etc.) in the text, in the ‘results’ section (lines 227-232)
Comment 4:Given the study's retrospective nature, establishing causality is challenging. Factors like physical activity, sleep, stress levels, and socioeconomic status, which influence both obesity and dietary choices, should be controlled or at least acknowledged. Without these, it is difficult to attribute increased BMI and adiposity solely to UPF consumption.
Reply to comment 4:We thank the reviewer for the advice. We have included information about the socio-economic status and physical activity levels of the study population in tables 1 and 4 and in the text (lines 203-210; lines 250-254).
Within the limits of the work we have made it explicit that no validated questionnaires were administered to assess the sleep quality and stress levels of the enrolled subjects (lines 386-388)
Comment 5: Although the study mentions several anthropometric and biochemical indices, it would be beneficial to specify which adiposity measurements were taken (e.g., waist-to-hip ratio, waist circumference, visceral fat percentage). A clear explanation of why specific measurements were chosen and how they relate to UPF intake would add depth to the analysis.
Reply to comment 5:We thank the auditor for the tip. We have added the requested information (lines 374-379) in the ‘Discussion’ section.
Comment 6: The study could strengthen its conclusions by discussing potential mechanisms by which UPF consumption might contribute to obesity and adiposity beyond caloric impact. For instance, the impact of additives, food matrix alterations, and hyper-palatability on satiety and metabolic health would provide a more comprehensive understanding of how UPFs exacerbate obesity.
Reply to comment 6: We apologize for the previous incomplete examination of the field of interest. We have expanded the discussion as requested (lines 326-329; lines 336-347; lines 354-357; lines 362-367)
All changes made to the original text, including newly added references, have been written in red
Reviewer 2 Report
Comments and Suggestions for Authors
Thank you for submitting the manuscript "A close relationship between ultra-processed foods and adiposity in Southern Italy adults" to Nutrients. The manuscript investigates the relationship between the consumption of ultra-processed foods and adiposity.
Although the experimental design was well conducted, the number of patients is very discrepant between the groups. In addition, no gold standard methods were used to contribute to the discussion (such as DEXA) and validate the results found by other methods. Thus, the results presented are still simple and the way they were presented seems to be the writing of a partial result of the research.
Comments on the Quality of English LanguageIt's ok.
Author Response
Comment 1: Although the experimental design was well conducted, the number of patients is very discrepant between the groups. In addition, no gold standard methods were used to contribute to the discussion (such as DEXA) and validate the results found by other methods. Thus, the results presented are still simple and the way they were presented seems to be the writing of a partial result of the research.
Reply to comment 1: We thank the reviewer for the suggestions and apologize for the shortcoming. We have specified that this is preliminary data (line 387) to be extended with a larger sample. We have also expanded the ‘results’ section (lines 203-210; lines 227-232; lines 250-254) to provide a deeper analysis and the ‘discussion’ section by adding the requested information (lines 374-379)
All changes made to the original text, including newly added references, have been written in red
Reviewer 3 Report
Comments and Suggestions for Authors
1) The authors should provide a detailed description of the protocol followed for Bioelectrical Impedance Analysis (BIA) measurements, including specific guidelines for subject preparation.
2) Similarly, the authors should clarify the protocol for blood pressure measurements, specifying which guidelines were followed and according to which criteria hypertension was diagnosed.
3) Lines 146-150: Replace the Italian terms for male and female with the corresponding English terms.
4) The authors should expand on the statistical analysis section. They should detail how data were presented, describe the method used to assess the normality assumption, and explain how they evaluated the homogeneity assumption.
5) Line 173: Correct the term "tot."
6) In Tables 1, 2, and 3, replace all instances of "ns" in the p-value column with the exact p-values.
7) In the tables, indicate the number of participants using the following format:
"All patients (n = 175)"
8) In Tables 1, 2, and 3, remove all commas preceding parentheses.
9) In Table 3, replace the word "die" with "day."
10) The first paragraph of the discussion section should clearly summarize the main findings of the study.
11) The authors should address additional limitations of their study, such as the retrospective and observational study design and the absence of multiple regression analyses to evaluate their research hypothesis; instead, only basic statistical tests (e.g., the ANOVA test) were performed.
12) The conclusion section should be revised to align with the study design of the current research. For example, the authors could state: "In conclusion, the results of the present study suggest that the consumption of UPFs may jeopardize human health."
Comments on the Quality of English LanguageThe manuscript needs moderate revision for language and grammar.
Author Response
Comment1: The authors should provide a detailed description of the protocol followed for Bioelectrical Impedance Analysis (BIA) measurements, including specific guidelines for subject preparation.
Reply to comment 1: As requested, we have made explicit the detailed protocol used for performing the bioimpedance analysis (lines 122-134)
Comment 2: Similarly, the authors should clarify the protocol for blood pressure measurements, specifying which guidelines were followed and according to which criteria hypertension was diagnosed.
Reply to comment 2: As requested, we have provided the detailed protocol for blood pressure assessment and reference guidelines for the diagnosis of hypertension (lines 140-152).
Comment 3: Lines 146-150: Replace the Italian terms for male and female with the corresponding English terms.
Reply to comment 3: We apologize for the error. We have made the appropriate corrections in the text (lines 182-186).
Comment 4: The authors should expand on the statistical analysis section. They should detail how data were presented, describe the method used to assess the normality assumption, and explain how they evaluated the homogeneity assumption.
Reply to comment 4: We thank the reviewer for the advice. We have expanded the ‘statistical analysis’ section by making it explicit that we used the Kolmogorov-Smirnoff test to assess the normal distribution of the data and the Levene's test to assess the equality of variance (lines 194-198).
Comment 5: Line 173: Correct the term "tot."
Reply to comment 5: As request we have made the appropriate corrections in the text (lines 222)
Comment 6: In Tables 1, 2, and 3, replace all instances of "ns" in the p-value column with the exact p-values.
Reply to comment 6: As request we have made the appropriate corrections in the tables.
Comment 7: In the tables, indicate the number of participants using the following format:
"All patients (n = 175)"
Reply to comment 7: We thank the reviewer for the advice. We have made the appropriate corrections in the tables.
Comment 8: In Tables 1, 2, and 3, remove all commas preceding parentheses.
Reply to comment 8: As request we have made the appropriate corrections in the tables.
Comment 9: In Table 3, replace the word "die" with "day."
Reply to comment 9: As request we have made the appropriate corrections in the tables.
Comment 10: The first paragraph of the discussion section should clearly summarize the main findings of the study.
Reply to comment 10: As requested, we have added a summary of the main results at the beginning of the ‘discussion’ section (lines 309-310)
Comment 11: The authors should address additional limitations of their study, such as the retrospective and observational study design and the absence of multiple regression analyses to evaluate their research hypothesis; instead, only basic statistical tests (e.g., the ANOVA test) were performed.
Reply to comment 11: We agree with the reviewer and thank him/her for the suggestion. We have implemented and further detailed the limits of our work in the specific section (lines 384-386)
Comment 12: The conclusion section should be revised to align with the study design of the current research. For example, the authors could state: "In conclusion, the results of the present study suggest that the consumption of UPFs may jeopardize human health."
Reply to comment 12: We thank the reviewer for the advice. We have shaped the conclusions as requested.
All changes made to the original text, including newly added references, have been written in red
Round 2
Reviewer 1 Report
Comments and Suggestions for Authors
After seeing the total modifications and additions made to improve the manuscript, I think it can be published.
Reviewer 2 Report
Comments and Suggestions for Authors
This reviewer thanks the authors for having made all the corrections requested by this reviewer and in this way, the quality of the manuscript was improved and therefore, my suggestion is to accept it for publication.
Reviewer 3 Report
Comments and Suggestions for Authors
The authors have addressed all my concerns
Comments on the Quality of English LanguageMinor English editing is required